# Histone and Histone Acetylation-Related Alterations of Gene Expression in Uninvolved Psoriatic Skin and Their Effects on Cell Proliferation, Differentiation, and Immune Responses

**DOI:** 10.3390/ijms241914551

**Published:** 2023-09-26

**Authors:** Dóra Romhányi, Kornélia Szabó, Lajos Kemény, Gergely Groma

**Affiliations:** 1Department of Dermatology and Allergology, University of Szeged, H-6720 Szeged, Hungary; romhanyidora9411@gmail.com (D.R.); szabo.kornelia@med.u-szeged.hu (K.S.); kemeny.lajos@med.u-szeged.hu (L.K.); 2Hungarian Centre of Excellence for Molecular Medicine-University of Szeged Skin Research Group (HCEMM-USZ Skin Research Group), H-6720 Szeged, Hungary; 3HUN-REN-SZTE Dermatological Research Group, H-6720 Szeged, Hungary

**Keywords:** psoriasis, uninvolved skin, histone, histone acetylation and deacetylation, proliferation, immune responses

## Abstract

Psoriasis is a chronic immune-mediated skin disease in which the symptom-free, uninvolved skin carries alterations in gene expression, serving as a basis for lesion formation. Histones and histone acetylation-related processes are key regulators of gene expression, controlling cell proliferation and immune responses. Dysregulation of these processes is likely to play an important role in the pathogenesis of psoriasis. To gain a complete overview of these potential alterations, we performed a meta-analysis of a psoriatic uninvolved skin dataset containing differentially expressed transcripts from nearly 300 individuals and screened for histones and histone acetylation-related molecules. We identified altered expression of the replication-dependent histones HIST2H2AA3 and HIST2H4A and the replication-independent histones H2AFY, H2AFZ, and H3F3A/B. Eight histone chaperones were also identified. Among the histone acetyltransferases, ELP3 and KAT5 and members of the ATAC, NSL, and SAGA acetyltransferase complexes are affected in uninvolved skin. Histone deacetylation-related alterations were found to affect eight HDACs and members of the NCOR/SMRT, NURD, SIN3, and SHIP HDAC complexes. In this article, we discuss how histone and histone acetylation-related expression changes may affect proliferation and differentiation, as well as innate, macrophage-mediated, and T cell-mediated pro- and anti-inflammatory responses, which are known to play a central role in the development of psoriasis.

## 1. Introduction

Psoriasis is an inflammatory skin disease with an exaggerated response to external and internal stress reactions, resulting in keratinocyte hyperproliferation, abnormal differentiation, and massive immune cell infiltration [1,2]. The combined interaction of abnormal genetic, epigenetic, environmental, and microbiome-related factors is believed to be responsible for the development of psoriasis [3]. In this disease, the macroscopically healthy, uninvolved skin carries multiple molecular changes that lead to the appearance of symptoms [4,5]. Large-scale analyses comparing healthy, uninvolved, and psoriatic skin samples have found that the expression levels of many genes differ [6]. Epigenetic changes related to histones through their post-translational modification are partly behind these processes [7].

Chromatin is composed of DNA and histones, of which two main types can be distinguished: the gene-poor, transcriptionally less active heterochromatin and the gene-rich euchromatin, which is accessible for transcription [8]. The basic unit of chromatin is the nucleosome, composed of DNA and a core histone octamer [9], while the higher-order chromatin structures are promoted by the H1 linker histone [10]. On the basis of their role in replication, replication-dependent canonical [11] and replication-independent non-canonical histone variants [12] have been distinguished, encoded by 75 and 20 genes, respectively [13].

There are four classes of histone chaperones. Class I contains single chaperones, class II is a multichaperone complex, class III is enzymatic, and class IV is a multiclass chaperone complex [14]. These chaperones regulate the assembly, deposition, removal, exchange, and transport of histones, thereby modulating proliferation [15] and inflammatory responses [16,17].

Histone acetylation, carried out by histone acetyltransferases (HATs), leads to transcriptional activation [18,19]. There are two major types of HATs, A- and B-type [20]. A-type HATs acetylate chromatin-incorporated histones, whereas B-type HATs acetylate newly synthesized histones [21,22]. By contrast, histone deacetylation by histone deacetylases (HDACs) results in transcriptional repression [23].

Regarding psoriasis pathogenesis, histone acetylation in general and H3K27Ac in particular show a different pattern in lesional skin compared with healthy skin, as seen in heat images [7]. Histone H3 acetylation plays a role in Th17 cell differentiation and keratinocyte proliferation, both of which are known to play a central role in the pathogenesis of psoriasis [24]. Elevated expression of the epigenetic modifier CREMα has been detected in psoriatic T cells [25], which has been suggested to be partially responsible for the development of the abnormal expression of IL2 and IL17 [26].

On the one hand, histone variants can replace and substitute each other. On the other hand, they differ in the number and position of post-transcription modification sites at their globular core and N-terminal tail, allowing them to carry out distinct and specialized roles, including the regulation of tissue- and cell-type-specific functions. These functions include the regulation of proliferation [27,28], cell fate commitment [29], hematopoiesis [30], differentiation [31], macrophage [32] and T cell immune responses, and mutagenesis of immunoglobins [33,34].

To gain insight into how these processes are affected in uninvolved psoriatic skin, we screened for gene expression alterations in histones, histone chaperones, and histone acetylation-related molecules. We used a psoriatic transcriptome dataset containing nearly 300 published individual patient data (99 psoriatic lesional, 27 uninvolved psoriatic, and 172 healthy samples) to determine how these alterations may affect key processes in the pathogenesis of this common skin disease, as well as proliferation and immune responses.

## 2. Results and Discussion

### 2.1. Histones

To the best of our knowledge, there are no studies on the role of histones regarding the development of psoriasis. Among the canonical histones, we found that H2AC18 (also known as HIST2H2AA3) and H4C14 (also known asHIST2H4A) showed different expression in psoriatic uninvolved skin compared with the skin of healthy individuals (Figure 1 and Appendix A).

In non-dividing cells, HIST2H2AA3 participates in the terminal differentiation program [31] (Figure 2). HIST2H4A is commonly used as a marker for proliferation [27,28] (Figure 2). Therefore, differential expression of these histones may contribute to proliferation and differentiation-related alterations in psoriasis.

Among the replication-independent histones that modulate nucleosome plasticity [12,35], MACROH2A1 (also known as H2AFY), H2AZ1 (also known as H2AFZ), and H3F3A/B show altered expression in uninvolved skin (Figure 1).

H2AFY plays a role in transcriptional repression [36] by regulating the transition between activating and inhibitory chromatin remodeling complexes [37]. It is also involved in the repression of pluripotent and bivalent developmental genes, thereby maintaining cell faith commitment [29] (Figure 2).

The H2AFY-PARP1 axis determines the cellular stress responses to DNA damage, heat shock, and aging [38]. H2AFY can suppress IFNB1 [39] and the proinflammatory cytokine IL-8 (CXCL8) [40], as well as CCL2 [41] transcription [42]. IFNB1 regulates the Th17 immune response [33], and IL-17A induces the production of IL-8 [43], while CCL2 promotes inflammatory processes in psoriasis [44]. Through the canonical JAK/STAT signaling pathway, the IFNB1-initiated response regulates cell proliferation [45], which is known to be dysregulated in psoriasis. Therefore, the differential expression of H2AFY in uninvolved psoriatic skin is likely to play a massive role in triggering psoriasis-related dysregulation in innate immune and proliferation-related responses (Figure 2).

H2AFZ ubiquitination regulates the transition between eu- and facultative heterochromatin, distinguishing constitutive from facultative heterochromatin [46] during the cell cycle (G1/S phase cMYC, Ki67) [47], and influencing lineage commitment [48,49] (Figure 2). Elevated expression of Ki67 and cMYC has been detected in psoriatic lesions, contributing to keratinocyte hyperproliferation [50]. Therefore, H2AFZ may contribute to the development of the disease by regulating stress response and proliferation, both of which are known to be involved in the pathogenesis of psoriasis.

H3F3A/B encodes histone H3.3, which is located at the euchromatin borders [51,52] and maintains heterochromatin structures [53]. As bifunctional histones, they can act as both transcriptional activators and repressors [54]. H3F3A/B is required in somatic cells to maintain their identity, for normal chromosome segregation [55], to maintain the balance of hematopoiesis [30], to activate macrophages [32], and to regulate the mutagenesis in the variable regions of immunoglobins [34] (Figure 2). In line with our results, an increased (hyper)mutation rate of IgE was detected in psoriasis patients [56].

In the development of abnormal differentiation, pluripotency, cell line commitment, and the differential expression of histones involved in terminal differentiation (H2AFY, H2AFZ, and HIST2H2AA3, respectively) may play a role during the development of the disease.

### 2.2. Histone Chaperones

Among the class I single chaperones, abnormal expression of the NPM1 and SET was identified in the non-lesioned skin (Figure 1).

NPM1 is an H1 and H3/H4 chaperone that participates in heterochromatin (re)arrangement [57]. NPM1 promotes cell proliferation [15] and is required for the maintenance of cell identity by maintaining a cell type-specific gene expression pattern [57]. The expression of NPM1 is elevated in proliferating keratinocytes of psoriatic lesions [58] and activates inflammatory responses when released into the extracellular space [16] (Figure 3).

The histone chaperone SET inhibits nucleosome acetylation and regulates p53-mediated cell cycle arrest [59]. SET regulates the G1/S and G2/M transition [60] via E-CDK2 and B-CDK1 [61], respectively (Figure 3), and inhibits cytotoxic T-cell-mediated apoptosis (www.genecards.org (accessed on 24 June 2023.)). These changes, characteristic of non-lesional skin, may be important in the development of the disease, as previous studies have shown increased activity of CDK1 and CDK2 in the psoriatic epidermis [62,63].

Among the class II. multi-chaperone complex members, CHAF1A, RBBP4, and UBN1 showed alterations in the non-lesioned skin (Figure 1).

CHAF1A is a component of the CAF1 complex that maintains Cd4 silencing in cytotoxic T cells [17]. The CAF1 complex is linked to DNA replication [64] and determines the proliferation–differentiation switch in stem cells [65], which is known to be abnormally regulated in psoriasis.

RBBP4 levels are upregulated in psoriasis by skin-derived mesenchymal stem cells, contributing to epidermal hyperplasia [66].

UBN1 is part of the bifunctional chaperone HIRA complex and participates in both transcriptional activation and inhibition [67]. By repressing proliferation-promoting genes, UBN1 regulates tissue aging-associated cellular senescence [68]. Consistent with our results, middle and upper epidermal keratinocytes of psoriatic plaques are characterized by a special state of aging, which is manifested by cell cycle arrest, as well as the release of inflammatory effectors and other molecules characteristic of aging [69].

Class III enzymatic complex members ANP32E and VPS72 also show altered expression. As part of the INO80 family, they regulate histone exchange [70]. ANP32E can remove H2AFZ from the nucleosome [71], while VPS72 deposits H2AFZ during mitosis [72], and immune cell infiltration [73,74] (Figure 3) that are known to be affected in the disease.

### 2.3. Histone Acetylation

Only type A HATs or their modulators showed abnormal expression in uninvolved psoriatic skin. Type A HATs can be classified into three subfamilies: the CBP/CREBBP, GNAT, and MYST families.

Members of the CBP/CREBBP family did not show transcriptional changes in uninvolved psoriatic skin. However, abnormal expression of EP300 modulators, such as the sequence-specific DNA binding protein MYBBP1A, the EP300 coactivator WBP2, and the EP300 corepressor CTBP1, was observed in the same samples (Figure 1). Elevated levels of CTBP1 have been demonstrated in psoriatic plaques, and mice overexpressing CTBP1 in epidermal keratinocytes show severe skin inflammation with increased expression of Th1 and Th17 cytokines [75], while WBP2 regulates epidermal [76] and T cell proliferation [77,78].

Abnormal expression of the GNAT family member ELP3 was also observed in uninvolved skin (Figure 1). ELP3 inhibits M1 and promotes M2 macrophage polarization [79].

We identified that the MYST family member KAT5 had altered expression in uninvolved skin (Figure 1). KAT5 modulates the differentiation and tissue infiltration of Th17 and Treg cells via FOXP3 [80]. As a cofactor of STAT3, KAT5 regulates IL-9 signaling [81] and hematopoietic stem cell maintenance [82] (Figure 4). KAT5 is also a catalytic subunit of the Tip60 histone acetyltransferase complex.

The H4 and H2A histone-specific acetyltransferase [83] and the lipid synthesis regulator [84,85] NAA40 are also differentially expressed in uninvolved psoriatic skin (Figure 1).

Type A histone acetyltransferases are components of several complexes that exert specific or universal effects [86]. As a result of their analysis, we found differential transcriptional expression of individual subunits of the NSL acetyltransferase complex and the SAGA deubiquitinase and histone acetyltransferase multicomplex with various transcription factor-interacting proteins [87,88], including TRRAP [88] (Figure 1).

The NSL complex regulates many mitochondrial processes, as well as transcription, RNA splicing, and telomere elongation [89]. As components of this complex, KANSL1 and MCRS1 show transcriptional alterations in uninvolved psoriatic skin (Figure 1).

KANSL1 is a master regulator of immune gene expression [90] (Figure 4), whereas MCRS1 protects chromosome-associated microtubules from depolymerization during mitosis [91] (Figure 4).

We identified a change in the expression of TADA2B in uninvolved skin. This is a part of the HAT module of the SAGA complex (Figure 1), which regulates p53 responses [92], stem cell pluripotency, and viability [93] (Figure 4).

TRRAP, which is responsible for recruiting transcription factors and histone acetyltransferases to chromatin, is required for transcriptional activation [94]. TRRAP [95] regulates the entry from the G0 to G1 phase and transitions between the different phases throughout the cell cycle [96], and by regulating critical differentiation markers, it maintains stem cells self-renewal and prevents their differentiation, both of which are known to be affected in psoriasis [97]. TRRAP represses the master regulator of interferon genes, IRF9 [95], whose expression is elevated in psoriasis [98]. TRRAP is also a component of the Tip60 complex, which promotes histone acetyltransferase activity [95]. Among Tip60 complex members, we identified abnormal expression of ACTB, BRD8, ING3, and KAT5 (discussed above) in uninvolved psoriatic skin (Figure 1). The TIP60 complex coactivators BRD8 and ING3 regulate p53-dependent gene suppression and the cell cycle [99,100] (Figure 4).

Members of the inhibitor of histone acetyltransferases (INHAT) complex ANP32A and SET inhibit p300/CBP (CREBBP)- and KAT2B (PCAF)-mediated histone acetylation [101] resulting in the silencing of HAT-dependent transcription. The SET protein (described above among histone chaperones) can inhibit histone H4 and H1 acetylation-dependent transcription [102].

The HAT module of the SAGA complex shares several components with the large acetyltransferase ATAC complex [103], which is one of the main regulators of mitosis through the acetylation of histone H3 and H4 [104]. The ATAC complex component MBIP shows altered expression in non-lesioned skin. Splice variations of this gene have been described in psoriasis [105] pathogenesis, in which they contribute to immune cell infiltration [106] and/or keratinocyte hyperproliferation.

### 2.4. Histone Deacetylation

Among the members of the HDACI histone deacetylase family, HDAC3 and HDAC8 showed altered expression in uninvolved skin (Figure 1). HDAC3 inhibition results in the reduced expression of AQP3 [107], contributing to skin dryness in uninvolved and lesional psoriatic skin [108] and a decrease in LPS-induced inflammatory gene expression in macrophages [109] (Figure 5A). HDAC3 is part of the NCOR/SMRT complex, which is responsible for nuclear receptor-mediated transcriptional repression [110]. From this complex, we observed the abnormal expression of the GPS2 and TBL1X genes (Figure 1).

GPS2 regulates proinflammatory cytokine production in macrophages [111] and inhibits proliferation by suppressing mitogen-activated protein kinase-mediated signaling [112].

TBL1X modulates Wnt/β-catenin and TNFA-regulated transcription [110] (Figure 5B). Elevated levels of TBL1X have been described previously in psoriasis [113].

Altered expression of HDAC8 in uninvolved skin may modulate (keratinocyte) tolerance to TLR2/6 ligand stimulation [114,115] and may increase T cell infiltration [116] (Figure 5A).

HDAC I family members HDAC1 and HDAC2 are normally expressed in uninvolved skin, but the expression of their repressor SPHK2 is altered (Figure 1). SPHK2 inhibits HDAC1/2 activity [117], thus altering the differentiation of Th17 cells in psoriasis [118].

In addition, we observed altered expression levels of several members of the HDAC1/2 protein complexes, which affect the function of NURD, SHIP, and SIN3 complexes (Figure 1).

The NURD complex is a multi-functional complex, playing a role in remodeling chromatin; regulating histone deacetylase activities; and controlling the development of T cells [119], their cell cycle progression, and progenitor cell maintenance [120]. The NURD complex contains an ATP-dependent CHD3/4 chromodomain helicase; a transcriptional repressor adaptor macromolecule GATAD2A; the histone tail and promoter-reading transcriptional coregulator MTA1; a histone-binding, chromatin-remodeling factor RBBP4; and a DNA-binding MBD2/3, which connects the complex with DNA methylation processes [121] (Figure 5B). Among these molecules, the expression of CHD4, GATAD2A, and MTA1 is altered in uninvolved skin (Figure 1).

CHD4 plays an important role in the early development of the basal epidermal layer and regulates the induction and development of hair follicles by destabilizing the interactions between DNA and histones [122]. In keratinocytes, CHD4 can increase tolerance to stress by limiting the expression of stress response genes [123]. CHD4 also regulates Th2 cell differentiation [124], CD8+ T-cell infiltration [125], and self-antigen expression in epithelial cells [126] (Figure 5B).

GATAD2A regulates proliferation [127] and naive pluripotency [128] in association with CHD4. MTA1 regulates the balance between hematopoietic cell renewal and differentiation [129] via the MyD88 pathway [130]. The overexpression of MTA1 triggers the downregulation of the macrophage-attracting chemokine receptor (CCR2) and ligands, leading to M2 polarization and impairing the cytotoxic effect of T cells, resulting in CD8+ T cell enrichment [131] (Figure 5B).

The HDAC1/2-containing SHIP complex exhibits DNA binding and chromatin remodeling capabilities [132]. We found that HSPA2, a member of the SHIP complex, exhibited altered expression in uninvolved psoriatic skin compared with the skin of healthy individuals (Figure 1). HSPA2 acts as a molecular chaperone and provides protection against the cytotoxic effects of heat shock [132], and its expression in keratinocytes increases with hypoxia [133]. This molecule contributes to early keratinocyte differentiation [134] and acts as an important factor in the establishment and maintenance of the properly layered epidermis [135] (Figure 5B).

The SIN3 multiprotein complex influences protein stability, transcriptional activity, aging and heterochromatinization events, cell proliferation/cell cycle progression, cell survival [136], and pluripotency maintenance [137]. Among the complex components, SIN3A showed abnormal expression in uninvolved skin (Figure 1). Sin3A regulated T cell development [138], in particular Th17 cell differentiation, and the establishment of their inflammatory potential [139]. While in the skin, the same molecule is known to regulate terminal differentiation and the maintenance of epidermis homeostasis [140] (Figure 5B).

Among the members of the HDACII family, HDAC4, HDAC5, and HDAC6 show altered expression in uninvolved skin (Figure 1).

The histone deacetylase HDAC4 acts as a transcriptional repressor, but it may exhibit both pro- and anti-inflammatory effects depending on the target gene. While HDAC4-induced NF-κB gene expression inhibition results in the decreased production of proinflammatory cytokines [141], when inflammatory processes are initiated, it can also increase inflammation by indirectly activating Foxo3a [142]. HDAC4 also inhibits keratinocyte proliferation [143] (Figure 5A).

On the one hand, the overexpression of HDAC5 contributes to the initiation of apoptotic processes in keratinocyte stem cells [143]. On the other hand, it also regulates the transformation of CD4+ T cells into Tregs and the cytokine production of CD8+ T cells [144]. Fluid shear stress stimulates the phosphorylation and nuclear export of HDAC5, which plays an important role in the establishment and maintenance of flow-regulated anti-inflammatory processes [145] (Figure 5A).

Another HDACII family member, HDAC6, promotes cell motility [146] during wound healing [147] and chemotaxis of T lymphocytes [148], and it regulates the organization of immune synapses [149] (Figure 5A).

Among the HDACIII family members, SIRT5 and SIRT6 showed abnormal expression in uninvolved skin (Figure 1).

SIRT5 negatively regulates keratinocyte proliferation and inflammation (TNFA induction [150]) and improves epidermal barrier dysfunction [151] (Figure 5A).

SIRT6-mediated histone H3 deacetylation at the N-terminal tail (H3K9Ac) and during the cell cycle at the globular core (H3K56Ac) regulates telomeric chromatin structure, which is necessary to maintain genomic stability and lifespan [152]. By contrast, SIRT6-mediated deacetylation at H3K18 of the pericentric chromatin prevents proliferation-related (replicative) cellular senescence [153]. Changes in SIRT6 expression were also reported in association with the adaptive immune responses [150]. It regulates the balance between the M1 and M2 macrophages, influences wound healing [154], inhibits skin inflammation [155], and plays a role in cDC differentiation and function [156] (Figure 5A).

The HDACIV family member HDAC11 was also differentially expressed in uninvolved skin (Figure 1). The biological function of this family is incomplete. HDAC11 plays an important role in immune regulation, neutrophil lineage commitment, and inflammatory responses, including the regulation of macrophage IL10 and IL1B secretion, dendritic cell IL1B secretion, and T cell activation [157] (Figure 5A).

## 3. Materials and Methods

### 3.1. Establishment of the Psoriatic Transcriptome Sequencing Data Set

The dataset we used for these investigations was successfully used in another study that screened for psoriasis-related alterations affecting the peripheral nervous system in psoriatic uninvolved and lesional skin [158]. The combined transcriptome sequencing data were obtained from three studies [159,160,161] that randomly enrolled individuals with chronic plaque-type psoriasis and healthy donors (number of samples: lesional psoriatic: 99, uninvolved psoriatic: 27, healthy: 172). Skin punch biopsies were collected with no gender or age (>18) preferences for RNA sequencing. Psoriatic patients (PASI: min. 1% of total body surface area) on topical and systemic anti-psoriatic treatments had a washout period (the time between the last treatment and sample collection intended to exclude the interference of medication-related effects) of 1 and 2 weeks, respectively, prior to biopsy collection in all studies.

### 3.2. RNA Sequencing, Data Processing, and Differential Expression Analysis

RNA sequencing data processing and analysis were performed as described previously [158]. Briefly, the three RNA sequencing datasets [159,160] (ID Accession numbers: SRP035988, SRP050971, and SRP055813) were downloaded from the Sequence Read Archive using SRA tools (v2.9.2), and all available samples were uniformly reprocessed. Transcript levels were quantified using Kallisto (version 0.43.0) [162] and full transcriptome annotation GENCODE [163] v27 software. Transcript-level, length-scaled TPM (Transcripts Per Million) expression estimates from Kallisto were imported into the R statistical environment (v3.4.3) using the tximport [164] package (v1.6.0). The data were TMM-normalized [165] (edgeR [166] v3.20.9) and voom-transformed (limma [167,168] v3.34.9). voomWithQualityWeights() was used to combine the observation-level weight of the transcripts with the sample-specific weight, retaining lower-quality samples but down-weighing them in the analysis. Differential expression between uninvolved and healthy sample groups was tested using Limma. A linear model was fitted (limma lmFit), and the moderated t-statistics were calculated (eBayes). Differentially expressed transcripts (DETs) were defined if they had an FDR [168,169] corrected *p*-value of <0.05.

### 3.3. Screening for Histones and Histone Acetylation-Related DETs

Differentially expressed transcripts (DETs) from the NL vs. H (non-lesional/uninvolved and healthy, respectively) comparison were analyzed using libraries of datasets downloaded from https://amigo.geneontology.org/amigo/term/ (accessed on 24–29 June 2023) and supplemented with literature data. A complete list of libraries is shown in Appendix A, in which the GO database and literature datasets [13,14,73,87,101,103,132,170,171,172,173] are listed separately. The filtering used to determine matches between NL vs. H and the downloaded dataset was performed in Python by applying intersection analysis. Detailed information on all methodological steps and processes of the study is provided in Appendix A.

## 4. Conclusions

On the basis of our findings, we identified complex expression abnormalities in histones and genes with functions in histone acetylation-related processes.

There are already some therapies available to alleviate the clinical manifestation of the symptoms, which are based on a significant number of genetic/epigenetic studies [174]. The regulatory effect of HDAC inhibitors on T cells has been reported. According to these studies, in the presence of histone deacetylase inhibitors, the release of Th1 cytokines and the polarization of Th17 cells decreases, while the formation of Treg cells increases [175]. In addition, HDAC inhibition also modulates pigmentation by reducing MITF expression [176]. A study on the pan-HDAC inhibitor vorinostat found that it induced the apoptosis and differentiation of keratinocytes—consistent with the inhibition of keratinocyte proliferation in psoriasis [177]. According to recent research, the HDAC1 inhibitor (entinostat) reduced the infiltration of IL-17A+ γδT cells into the skin [178].

The pan-BET bromodomain HAT inhibitor (JQ1) reduced the ratio of IL17A+/IFNY+ T cells and IL17A secretion in both psoriatic arthritis patients and healthy individuals [179]. The CREBBP and P300-specific (type A HATs) inhibitor (CBP30) reduced the induced Th17 response in patients with psoriatic arthritis [180].

The described alterations are likely to contribute to the dysregulation of proliferation and differentiation, pro- and anti-inflammatory processes mediated by innate and professional immune cells in uninvolved psoriatic skin, leading to disease flare-ups. Further experimental confirmation of their functional modification may represent new points of intervention.

## 5. Limitations of the Study

It is important to note that no fold change cut-off was used as a criterion for differential expression; therefore, minor differences between uninvolved and healthy skin (FDR < 0.05) are also included in the study. These minor differences (as well as all others) in the expression were observed at the level of RNS transcripts, some of which may not have manifested at the protein level due to post-transcriptional, translational, and post-translational events, including the processing and degradation of proteins. In addition, skin biopsies contain both the epidermis and the dermis. These two layers of the skin contain different cell types, including keratinocytes, melanocytes, Merkel cells, fibroblasts, and several resident immune cells, like T cells, dendritic cells, Langerhans cells, NK cells, and macrophages. Therefore, the cell type in which the mRNA expression differences manifest could not be determined with certainty, and further experimental confirmation is required to support them, for which these results provide a strong basis.

## Figures and Tables

**Figure 1 ijms-24-14551-f001:**
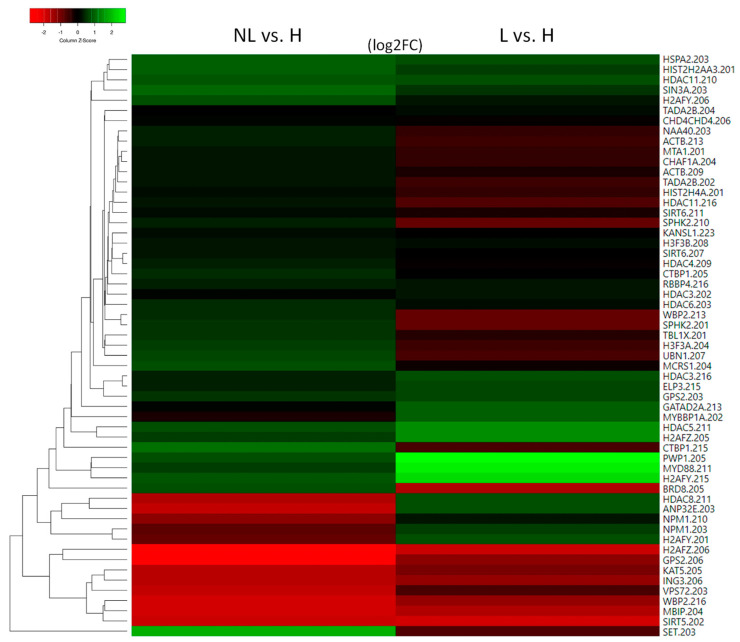
Heatmap of histone and histone acetylation-related molecules with altered expression in uninvolved psoriatic skin (NL) and their expression in lesional skin (L) compared with healthy skin (H).

**Figure 2 ijms-24-14551-f002:**
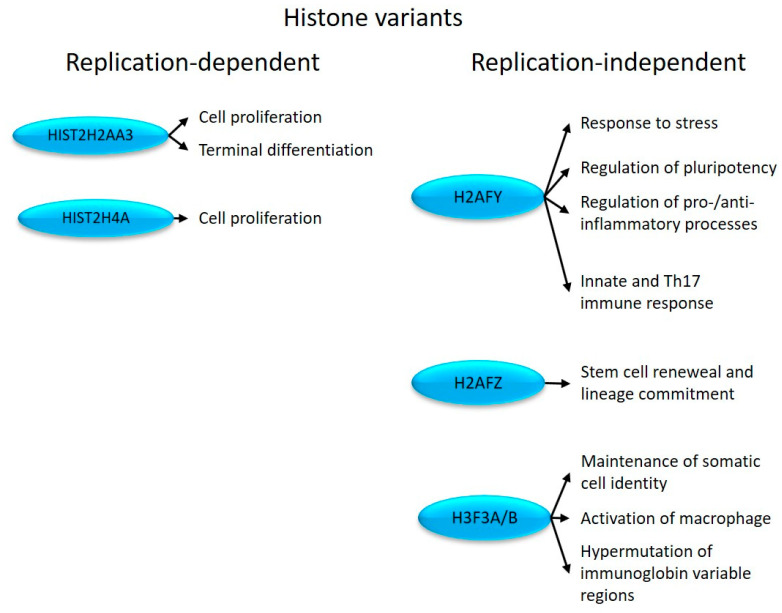
Replication-dependent and -independent histones with altered expression in psoriatic uninvolved skin and their effects on cell proliferation and immune system-related processes.

**Figure 3 ijms-24-14551-f003:**
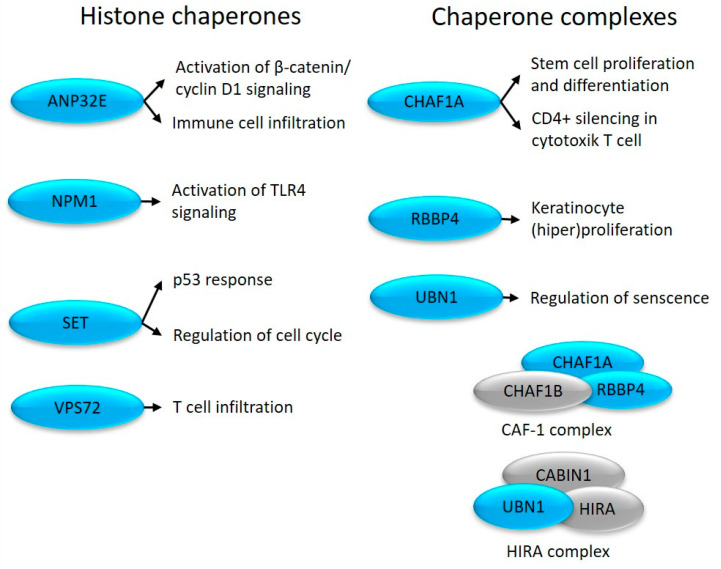
Altered expression of histone chaperones in uninvolved skin (depicted in blue) and their role in cell proliferation and immune system-related processes.

**Figure 4 ijms-24-14551-f004:**
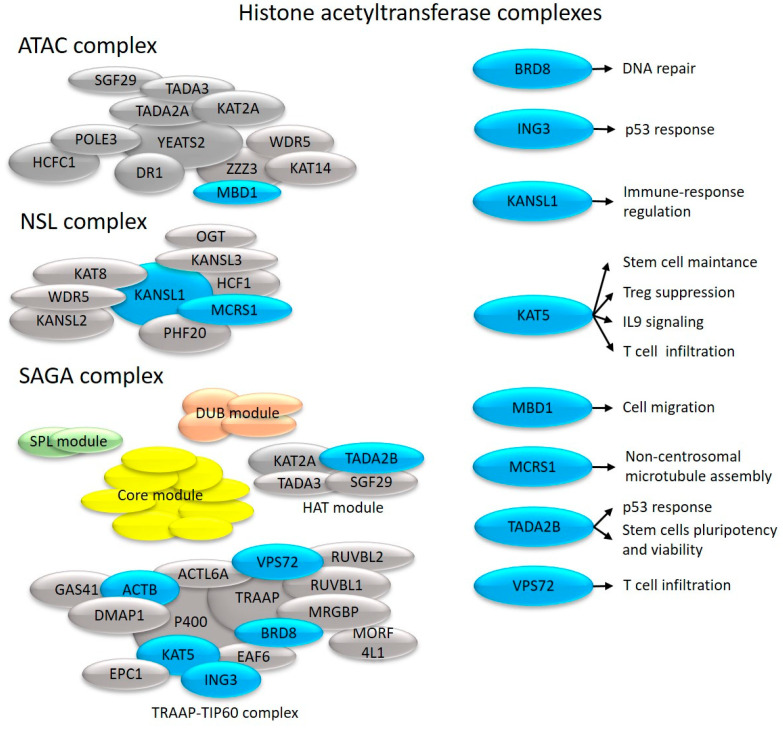
Effects on cell proliferation and immune system-related processes of histone acetyltransferase complex components with altered transcription in uninvolved skin (depicted in blue).

**Figure 5 ijms-24-14551-f005:**
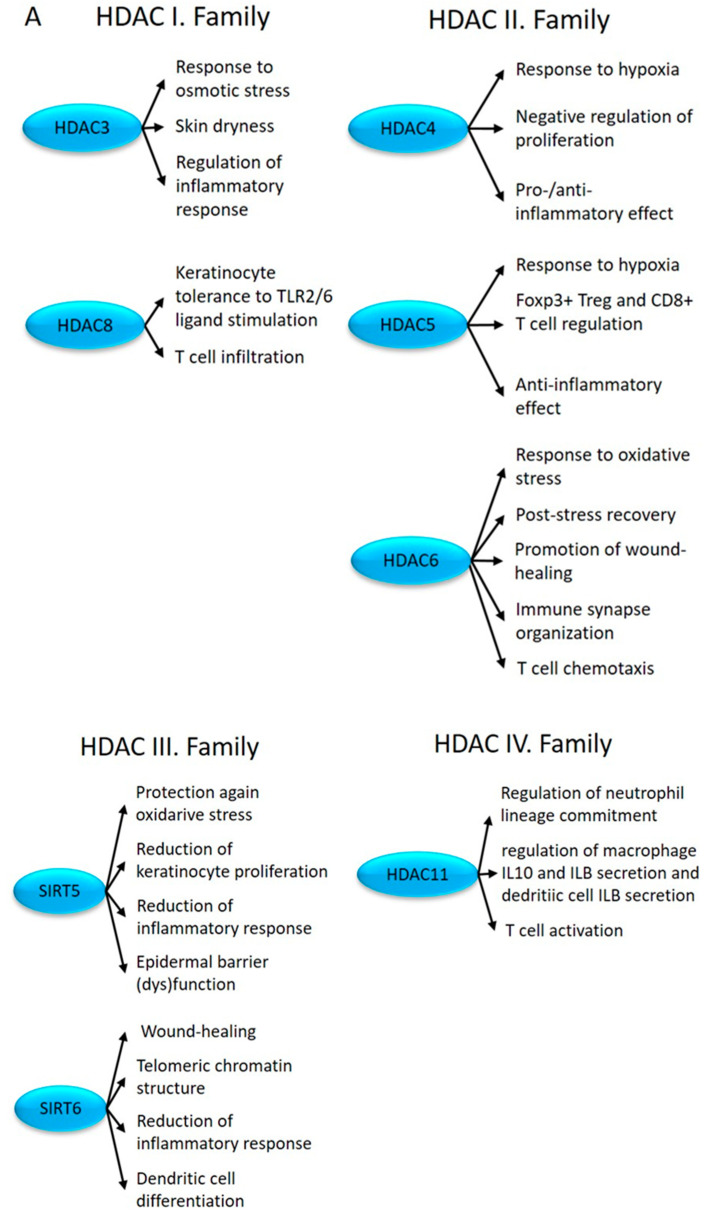
The impact of differentially expressed HDACs (**A**) and HDAC complexes (**B**) (depicted in blue) on proliferation, differentiation, and immune regulation in uninvolved skin.

## Data Availability

Only publicly available data was used in the study (Sequence Read Archive, https://www.ncbi.nlm.nih.gov/sra (accessed on 15 November 2021); study ID: SRP035988, SRP050971, and SRP055813).

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
