# Peer review of "Histone and Histone Acetylation-Related Alterations of Gene Expression in Uninvolved Psoriatic Skin and Their Effects on Cell Proliferation, Differentiation, and Immune Responses"

_ijms, 2023, doi:10.3390/ijms241914551_

Round 1

Reviewer 1 Report

Dear authors

I read your manuscript concerning the histone and histone acetylation-related gene expression alterations in psoriasis. The subject is interesting and the paper presents an overview of the current findings in a database. However, some points must be addressed and clarified.

1)     The introduction starts in medias res. You should report some lines about psoriasis as a complex disease and then report the molecular pathways implicated. To improve your manuscript in the introduction section, read and cite:

-        Caputo, V., Strafella, C., Cosio, T., Lanna, C., Campione, E., Novelli, G., Giardina, E., & Cascella, R. (2021). Pharmacogenomics: An Update on Biologics and Small-Molecule Drugs in the Treatment of Psoriasis. Genes12(9), 1398. https://doi.org/10.3390/genes12091398

-        Jiang, Y., Lu, S., Lai, Y., & Wang, L. (2023). Topical histone deacetylase 1 inhibitor Entinostat ameliorates psoriasiform dermatitis through suppression of IL-17A response. Journal of dermatological science110(3), 89–98. https://doi.org/10.1016/j.jdermsci.2023.05.001

-        Caputo, V., Strafella, C., Termine, A., Dattola, A., Mazzilli, S., Lanna, C., Cosio, T., Campione, E., Novelli, G., Giardina, E., & Cascella, R. (2020). Overview of the molecular determinants contributing to the expression of Psoriasis and Psoriatic Arthritis phenotypes. Journal of cellular and molecular medicine24(23), 13554–13563. https://doi.org/10.1111/jcmm.15742

-        Carlsson, E., Cowell-McGlory, T., & Hedrich, C. M. (2023). cAMP responsive element modulator α promotes effector T cells in systemic autoimmune diseases. Immunology, 10.1111/imm.13680. Advance online publication. https://doi.org/10.1111/imm.13680

2)     A major concern in the paper is using one database, while other publications or research in other databases have been excluded. In this view, study limitations are missing. Add this section at the end of the discussion.

3)     You should provide some lines about the clinical implication of your finding and the possible application in practice.

4)     Acknowledgments section must be modified.

5)     You should check all the references: there are duplicates and no APA style in the citation.

6)     The main files should be more fluent. Some sections are formally correct, but they look disconnected from the others. 

Moderate editing of English language required

Author Response

Dear Reviewer,

Thank you for your kind comments and suggestions, based on which we have revised our manuscript. All modifications are highlighted in yellow in the text. We hope that you will find the improved manuscript acceptable for publishing. Below, please find our responses to your comments and observations.

Kind regards,

Gergely Groma

1)The introduction starts in medias res. You should report some lines about psoriasis as a complex disease and then report the molecular pathways implicated. To improve your manuscript in the introduction section, read and cite:

-Caputo, V., Strafella, C., Cosio, T., Lanna, C., Campione, E., Novelli, G., Giardina, E., & Cascella, R. (2021). Pharmacogenomics: An Update on Biologics and Small-Molecule Drugs in the Treatment of Psoriasis. Genes, 12(9), 1398. https://doi.org/10.3390/genes12091398

-Jiang, Y., Lu, S., Lai, Y., & Wang, L. (2023). Topical histone deacetylase 1 inhibitor Entinostat ameliorates psoriasiform dermatitis through suppression of IL-17A response. Journal of dermatological science, 110(3), 89–98. https://doi.org/10.1016/j.jdermsci.2023.05.001

-Caputo, V., Strafella, C., Termine, A., Dattola, A., Mazzilli, S., Lanna, C., Cosio, T., Campione, E., Novelli, G., Giardina, E., & Cascella, R. (2020). Overview of the molecular determinants contributing to the expression of Psoriasis and Psoriatic Arthritis phenotypes. Journal of cellular and molecular medicine, 24(23), 13554–13563. https://doi.org/10.1111/jcmm.15742

-Carlsson, E., Cowell-McGlory, T., & Hedrich, C. M. (2023). cAMP responsive element modulator α promotes effector T cells in systemic autoimmune diseases. Immunology, 10.1111/imm.13680. Advance online publication. https://doi.org/10.1111/imm.13680

Thank you for your kind suggestions. The introduction has been modified according to your observations, and now also includes the highly relevant suggested papers as references.

2) A major concern in the paper is using one database, while other publications or research in other databases have been excluded. In this view, study limitations are missing. Add this section at the end of the discussion.

Thank you for your kind comment, indeed the description of our methodology was not detailed enough. Following your suggestions, we have provided a detailed description of our screening strategy methodology, presented as supplementary methodological figure 1.

The strategy of literature mining to confirm GO term association of genes in literature, and to identify function(s) of GO terms associated and literature supplemented genes was performed as described previously (ref). In brief, to identify function(s) of differentially expressed transcripts (uninvolved vs. healthy skin) associated with literature supplemented GO term (listed below) datasets (supplementary table 1.) literature mining was carried out using the HGNC (HUGO Gene Nomenclature Committee) gene symbol(s) applying the following strategy: after downloading the GO term associated gene lists (Human, https://amigo.geneontology.org/amigo/term) by applying Geneontology ID numbers (listed below), first each downloaded gene name was searched together with a given GO term as keywords to confirm GO described association in literature using PubMed (https://pubmed.ncbi.nlm.nih.gov/) online databases. Output publications were checked manually within the manuscript, as well as in Genecards human gene database (https://www.genecards.org) to confirm association of each gene with the GO term described mechanism.

To supplement downloaded GO datasets with additional associated genes from literature, Data of the AMIGO database was compared with the PubMed (https://pubmed.ncbi.nlm.nih.gov/) literature database using keyword-based automated literature screen. Output publications were checked manually within the manuscript, as well as in Genecards database for additional genes related to the given GO term described process/mechanism. If AMIGO database was found to be incomplete after a manual check based on the literature data, we combined the two results, and included the literature-supported relevant data. These literary additions are listed in a separate tab in Supplementary Table 1, together with the corresponding AMIGO databases.

In the case of histones, the AMIGO GO term “Histone(s)” was not available, so after a keyword-based automated literature screen, we created the histone database based on literature data. This was based on an article containing the classification of histones, which, to the best of our knowledge, contains all currently known human histone variants. (Stefano Amatori et al. The dark side of histones: genomic organization and role of oncohistones in cancer; 2021.).

These datasets were then combined given rise to the library of histones, histone chaperones and histone (de)acetylation related genes. This library and the transcriptome database of differentially expressed transcripts of uninvolved psoriatic vs. healthy skin, (created by combining three published transcriptome analysis datasets,) was filtered to determine matches between the two datasets. The output of this filtering was a library of histones, histone chaperones and histone (de)acetylation related genes differentially expressed in uninvolved psoriatic skin vs. healthy. This library served as an input for a third literature screen together with given mechanism(s) term(s) (listed below) as keyword(s) to identify proliferation, differentiation and immune response related functions of (gene) dataset components. For this screen in PubMed the following keywords (listed in alphabetical order) were applied: Cell cycle, Dendritic cell, Differentiation, Epidermis, Immune, Immune cell, Innate immune, Inflammation, Hematopoietic and Hematopoiesis, Keratinocyte, Macrophage, Neutrophil, Pluripotency, Proliferation, Psoriasis, Self-renewal, Senescence, Skin, Stem cell, T cell.

All described screens and filtering(s) were conducted in a case insensitive manner (both upper- and lower-case letters were considered) and alternative names/aliases of genes were taken into account.

  1. Histone chaperone activity, Geneontology ID#: GO:0140713

Database used for literature search: Pubmed (https://pubmed.ncbi.nlm.nih.gov/)

Keywords used for literature search to supplement Geneontology dataset: Histone chaperone, Histone chaperone complex

Applied literature for supplementation: Assala Lamaa et al. Integrated analysis of H2A.Z isoforms function reveals a complex interplay in gene regulation, 2020; Dan Filipescu Developmental et al. roles of histone H3 variants and their chaperones, 2013; Daniel Moreno-Andrés et al. VPS72/YL1-Mediated H2A.Z Deposition Is Required for Nuclear Reassembly after Mitosis, 2020; Leanne De Koning et al. Histone chaperones: an escort network regulating histone traffic, 2007

  1. Histone acetylation, Geneontology ID#: GO:0016573

Database used for literature search: Pubmed (https://pubmed.ncbi.nlm.nih.gov/)

Keywords used for literature search to supplement Geneontology dataset: Histone acetylation, Histone acetyltransferase, Histone acetyltransferase complex

Applied literature for supplementation: Dominik A et al. Herbst Structure of the human SAGA coactivator complex, 2021; Liliana Arede et al. Buffering noise: KAT2A modular contributions to stabilization of transcription and cell identity in cancer and development 2020; Qilian Yang et al. Epigenetics in ovarian cancer: premise, properties, and perspectives, 2018; Sang-beom Seo et al. Regulation of Histone Acetylation and Transcription by Nuclear Protein pp32, a Subunit of the INHAT Complex, 2002; Vincenzo Di Cerbo et al. Cancers with wrong HATs: the impact of acetylation, 2013, Zhi Fang et al. The Role of Histone Protein Acetylation in Regulating Endothelial Function, 2021,

  1. Histone deacetylation, Geneontology ID#: GO:0016575

Database used for literature search: Pubmed (https://pubmed.ncbi.nlm.nih.gov/)

Keywords used for literature search to supplement Geneontology dataset: Histone deacetylase, Histone deacetylase complex, Histone deacetylation

Applied literature for supplementation: Epigenetics in ovarian cancer: premise, properties, and perspectives, 2018; Zhi Fang et al. The Role of Histone Protein Acetylation in Regulating Endothelial Function, 2021. Qilian Yang

Because of these steps, it is unlikely that genes that are currently included in any another database are missed. At the same time, we cannot rule out human errors during the manual confirmation steps. Therefore, we do not consider our screening strategy as a limitation. However, as you kindly suggested, a separate section discussing (other) limitations of the study are now included in the text.

3) You should provide some lines about the clinical implication of your finding and the possible application in practice.

We have included a paragraph on the potential clinical implications of our findings, which is now presented in the Conclusion section.

4) Acknowledgments section must be modified.

Thank you for your observation. Acknowledgments section is now removed from the manuscript.

5) You should check all the references: there are duplicates and no APA style in the citation.

Thank you for your observation; citations were checked and corrected.

6) The main files should be more fluent. Some sections are formally correct, but they look disconnected from the others.

As kindly suggested the text of the main files has been improved and has undergone extensive English language editing.

Reviewer 2 Report

Comments on the MS “Histone and histone acetylation-related alterations of gene expression in uninvolved psoriatic skin and their effects on cell proliferation, differentiation and immune responses” by Dóra Romhány et al.

Major Concerns

-        A schematic diagram with all the steps’ of study would be welcomed by readers.

-        The abstract needs to be updated: it seems to use terms interchangeably without prior introduction or clarification, potentially confusing the reader. For instance, "uninvolved skin" may not be immediately clear to those less familiar with the specific context of psoriasis. Statements like "suggest an imbalanced regulation" are vague. The abstract should clarify how the findings lead to this conclusion and what the potential implications might be. The authors claim alterations in processes controlling "cell proliferation and immune responses, among others." This 'among others' is vague and could benefit from elaboration or specification. more concise presentation of the methodology, clearer definitions, and a more direct interpretation of findings are recommended

-        The results section is dense with technical terms and abbreviations, making it inaccessible to a broader audience. For instance, terms like "HIST2H2AA3/H2AC18" and "H2AFY/MACROH2A1" without sufficient context can be overwhelming for readers.

-        The statement "show different expression" (line 74) is vague. It would be helpful to specify whether the expression is upregulated or downregulated in the context of the psoriatic uninvolved skin versus healthy skin.

-        The manuscript lists various roles and mechanisms associated with the mentioned histones. However, it fails to link these roles with the implications in the context of psoriasis clearly. Revise it.

-        Phrases like "is likely to play a massive role" (line 96) are not only broad but might also be considered speculative. It would be preferable to present specific evidence from the study that supports such claims.

-        While the chaperones are discussed, the significance of their altered expression, especially in the context of psoriatic skin, is not always evident. It would benefit from a more in-depth discussion on the implications of these alterations. Revise it.

-        Some parts of the section Histone Acetylation are too dense, making it difficult for readers to extract key information or follow the logic. The flow between histone chaperones, their functions, and the associated findings is not always straightforward. The connection between the different subfamilies of Type A HATs and their relevance to the primary topic seems fragmented. The content would benefit from a clearer, more structured presentation. TRRAP's discussion stretches from its cell cycle regulation to its involvement in the Tip60 complex. While detailed, the significance of this deep dive into TRRAP's roles remains unclear in the context of the broader topic. Revise it.

-        section "4. Materials and Methods" is generally well-structured, but the introduction to the section could provide a more comprehensive understanding of the broader goals of these methods.

-        In section 4.1, the authors mention that the dataset has been "successfully used in other studies." It would be helpful to provide more context about how and in what capacity the dataset was successfully utilized in prior studies.

-        There's a need to ensure the ethical approval of the datasets or mention the original sources' ethics statement, especially when dealing with human subjects. Although there is mention of "chronic plaque-type psoriasis and healthy donors", it would be beneficial to provide a more precise number of patients and controls included in the study. This can help readers understand the sample size and potential limitations.

-        The authors did a commendable job of including software versions, which is crucial for reproducibility. It would be helpful to ensure that all essential software and toolkits, including smaller scripts or libraries used, are adequately mentioned and referenced.

-        In section 4.3, while the data sources are clearly mentioned, there is limited detail on how "literature data" was supplemented. Was it manually curated? If so, what criteria were used?

-        The differential expression criteria in section 4.2 is clear (FDR corrected p-value < 0.05 and absolute log2 fold-change greater than 1). Still, a brief rationale for choosing these thresholds might help the readers understand the significance and any potential implications of these cutoffs.

-        It would be helpful to briefly define or explain certain technical terms such as "washout period", "TMM normalized", "voom transformed", etc., for readers unfamiliar with these terms.

-        There's a minor typo in the last line. It should be "Python" instead of "Phyton".

-        What perspectives for humans does this MS have?

-        Mention limitations.

-         Consider revision accordingly!

English minor revision 

Author Response

Dear Reviewer,

Thank you for your kind comments and suggestions, based on which we have thoroughly revised our manuscript. All modifications are highlighted in yellow in the text. We hope that you will find the improved manuscript acceptable for publication. Below, please find our responses to your comments and observations.

Kind regards,

Gergely Groma

Major Concerns

-A schematic diagram with all the steps of study would be welcomed by readers.

We have provided detailed information in supplementary figure 1. that includes and describes all the steps of our study.

-The abstract needs to be updated: it seems to use terms interchangeably without prior introduction or clarification, potentially confusing the reader. For instance, "uninvolved skin" may not be immediately clear to those less familiar with the specific context of psoriasis. Statements like "suggest an imbalanced regulation" are vague. The abstract should clarify how the findings lead to this conclusion and what the potential implications might be. The authors claim alterations in processes controlling "cell proliferation and immune responses, among others." This 'among others' is vague and could benefit from elaboration or specification.

The abstract has been updated according to your suggestions. The term 'among others' has been removed, due to lack of space in this section for an elaborate specification of all possible cellular mechanisms in question.

-More concise presentation of the methodology, clearer definitions, and a more direct interpretation of findings are recommended.

We have provided a supplementary figure describing in detail the methodology applied as well as clarifying the definitions. Regarding the interpretation of the results, we have now included a section describing the limitations of our study. In this, we emphasized that experimental validation is required to confirm our results, similar to other studies that based on the analysis of large data sets.

-The results section is dense with technical terms and abbreviations, making it inaccessible to a broader audience. For instance, terms like "HIST2H2AA3/H2AC18" and "H2AFY/MACROH2A1" without sufficient context can be overwhelming for readers.

As kindly suggested, terms such as "HIST2H2AA3/H2AC18" and "H2AFY/MACROH2A1 have been modified in the text, and technical and other abbreviations are explained at the end of the manuscript (List of abbreviations).

- The statement "show different expression" (line 74) is vague. It would be helpful to specify whether the expression is upregulated or downregulated in the context of the psoriatic uninvolved skin versus healthy skin.

In the study transcriptomic differences were analyzed (all different transcripts/splice variants of a given protein coding gene), which do not necessarily correlate with the amounts of proteins encoded by them due to modifying mechanisms following transcription (e.g., protein degradation). Moreover, some transcripts coded by same gene could have similar or opposite direction of relative change when compared to healthy. Therefore, if possible, we would like to keep the term differences to prevent possible misinterpretations. However, relative expressions are shown in the heatmap, and a complete list of differentially expressed transcripts and their relative expression level (log2fold change) are listed as supplement.

-The manuscript lists various roles and mechanisms associated with the mentioned histones. However, it fails to link these roles with the implications in the context of psoriasis clearly. Revise it.

Based on your comments, relevant modifications are now included in the text, regarding the possible role of histones in the context of psoriasis pathogenesis.

-Phrases like "is likely to play a massive role" (line 96) are not only broad but might also be considered speculative. It would be preferable to present specific evidence from the study that supports such claims.

We tried to modified our statements and support them with adequate references according to your kind proposal. The sequence and logic behind our all statements is that there are some expressional alterations of a given molecule, such molecules have known functions, and disturbance of these functions could (at least in part) contribute or explain disease associated altered mechanisms. Therefore, in some cases stronger statements cannot be made especially in case of a heterogeneous disease as we refer to at the limitation section. It is important to note that in this regard all findings of such studies (including transcriptomics, proteomics and metabolomics) could be considered speculative, due to the lack of supporting experiments, and only provide a base for them.

-While the chaperones are discussed, the significance of their altered expression, especially in the context of psoriatic skin, is not always evident. It would benefit from a more in-depth discussion on the implications of these alterations. Revise it.

The section discussing chaperones has been revised according to your kind suggestions.

-Some parts of the section Histone Acetylation are too dense, making it difficult for readers to extract key information or follow the logic. The flow between histone chaperones, their functions, and the associated findings is not always straightforward. The connection between the different subfamilies of Type A HATs and their relevance to the primary topic seems fragmented. The content would benefit from a clearer, more structured presentation.

The sections have been revised and modified according to your kind suggestions. Modifications are highlighted in yellow in the text.

TRRAP's discussion stretches from its cell cycle regulation to its involvement in the Tip60 complex. While detailed, the significance of this deep dive into TRRAP's roles remains unclear in the context of the broader topic. Revise it.

The section discussing the role of TRRAP has been modified according to your kind suggestions and contains detailed information in the context of psoriasis.

-Section "4. Materials and Methods" is generally well-structured, but the introduction to the section could provide a more comprehensive understanding of the broader goals of these methods.

We now provide detailed information on the broader aims and objectives, together with the applied methodological steps in supplementary figure 1.

-In section 4.1, the authors mention that the dataset has been "successfully used in other studies." It would be helpful to provide more context about how and in what capacity the dataset was successfully utilized in prior studies.

The sentence in question has been modified as follows: The dataset we used for these investigations has been successfully used in another study that investigating psoriasis-related alterations affecting the peripheral nervous system in the psoriatic uninvolved and lesional skin.

-There's a need to ensure the ethical approval of the datasets or mention the original sources' ethics statement, especially when dealing with human subjects.

Ethical approvals of individuals are presented in the original three studies from which our database was created from. Since we are only use data from these studies and did not include additional human samples, we do not address ethical approvals. However, if you still feel that this part is necessary, we could include the following sentence: Statements regarding ethical issues are provided in the three original publications from which data of human samples were used.

-Although there is mention of "chronic plaque-type psoriasis and healthy donors", it would be beneficial to provide a more precise number of patients and controls included in the study. This can help readers understand the sample size and potential limitations.

We have now provided detailed information regarding the number of patients and controls included in the study as part of the Materials and Methods section.

-The authors did a commendable job of including software versions, which is crucial for reproducibility. It would be helpful to ensure that all essential software and toolkits, including smaller scripts or libraries used, are adequately mentioned and referenced.

We have supplemented (or highlighted) references regarding all software implications. We also provide a reference of a similar study where the methods in question are discussed in depth, together with the schematic diagram in Supplementary Figure 1. We believe that the provided information is now sufficient to reproduce the study or perform similar analyses.

-In section 4.3, while the data sources are clearly mentioned, there is limited detail on how "literature data" was supplemented. Was it manually curated? If so, what criteria were used?

In order to provide more detailed information, a supplementary figure with a detailed figure legend is now provided, which includes and briefly explains all the steps of our study.

-The differential expression criteria in section 4.2 is clear (FDR corrected p-value < 0.05 and absolute log2 fold-change greater than 1). Still, a brief rationale for choosing these thresholds might help the readers understand the significance and any potential implications of these cutoffs.

We are particularly grateful for this question, as it pointed out a mistake in the manuscript. We did not apply any thresholds in the study, in order to describe all (including minor) significant differences in expression. Therefore, “absolute log2 fold-change greater than 1” was removed from the manuscript.

-It would be helpful to briefly define or explain certain technical terms such as "washout period", "TMM normalized", "voom transformed", etc., for readers unfamiliar with these terms.

As kindly suggested, brief definitions of the suggested terms are now included in the text. In the case of complex terms that cannot be defined briefly, appropriate references are provided (highlighted in yellow).

-There's a minor typo in the last line. It should be "Python" instead of "Phyton".

Thank you for your observation; the typo was corrected.

-What perspectives for humans do this MS have?

In addition to the different histone acetylation patterns observed in uninvolved psoriatic skin, there is increasing evidence suggesting alterations of these processes in lesions. A detailed understanding of the role of histones/chaperones/acetylases/deacetylases in the pathomechanism of psoriasis may provide novel intervention points for the management of the disease by modulating the functions of HDACs/HAT. We have included a paragraph on the potential clinical implications of our findings, and it is now presented in the Conclusion section.

-Mention limitations.

As you kindly suggested, a separate section discussing limitations of the study are now included in the text.

Comments on the Quality of English Language: English minor revision

The text of the main files has been improved and has undergone extensive English language editing.

Round 2

Reviewer 1 Report

Dear Authors,

all the corrections have been made and the manuscript has been improved. 

Author Response

Dear Reviewer,

Thank you very much for your time, kind comments, suggestions and remarks, which contributed greatly to the improvement of the manuscript. We observed and corrected a mistake in the manuscript (corrected line: 64, highlighted in blue).

Kind Regards,

Gergely Groma

Reviewer 2 Report

The authors were unable to incorporate all the recommended revisions and suggestions into the manuscript.

minor corrections needed

Author Response

Dear Reviewer,

Thank you very much for your time, kind comments, suggestions and remarks, which contributed greatly to the improvement of the manuscript. We observed and corrected a mistake in the manuscript (corrected line: 64, highlighted in blue).

We hope that you will find the revised manuscript suitable for publication. However, if you feel that any further specific tasks or modifications are needed in the manuscript, please do not hesitate to name them.

Kind Regards,

Gergely Groma
